# Novel Unspecific Peroxygenase from *Truncatella angustata* Catalyzes the Synthesis of Bioactive Lipid Mediators

**DOI:** 10.3390/microorganisms10071267

**Published:** 2022-06-22

**Authors:** Rosalie König, Jan Kiebist, Johannes Kalmbach, Robert Herzog, Kai-Uwe Schmidtke, Harald Kellner, René Ullrich, Nico Jehmlich, Martin Hofrichter, Katrin Scheibner

**Affiliations:** 1Institute of Biotechnology, Brandenburg University of Technology Cottbus-Senftenberg, Universitätsplatz 1, 01968 Senftenberg, Germany; jan.kiebist@izi-bb.fraunhofer.de (J.K.); johannes.kalmbach@b-tu.de (J.K.); kai-uwe.schmidtke@b-tu.de (K.-U.S.); 2Fraunhofer Institute for Cell Therapy and Immunology, Branch Bioanalytics and Bioprocesses IZI-BB, Am Mühlenberg 13, 14476 Potsdam, Germany; 3Unit of Environmental Biotechnology, TU Dresden—International Institute Zittau, Markt 23, 02763 Zittau, Germany; robert.herzog1@tu-dresden.de (R.H.); harald.kellner@tu-dresden.de (H.K.); rene.ullrich@tu-dresden.de (R.U.); martin.hofrichter@tu-dresden.de (M.H.); 4Department of Molecular Systems Biology, Helmholtz-Centre for Environmental Research, UFZ, Permoserstrasse 15, 04318 Leipzig, Germany; nico.jehmlich@ufz.de

**Keywords:** eicosanoids, lipid mediators, EETs, HETEs, unspecific peroxygenases, human drug metabolites, biocatalysis, *Tan*UPO

## Abstract

Lipid mediators, such as epoxidized or hydroxylated eicosanoids (EETs, HETEs) of arachidonic acid (AA), are important signaling molecules and play diverse roles at different physiological and pathophysiological levels. The EETs and HETEs formed by the cytochrome P450 enzymes are still not fully explored, but show interesting anti-inflammatory properties, which make them attractive as potential therapeutic target or even as therapeutic agents. Conventional methods of chemical synthesis require several steps and complex separation techniques and lead only to low yields. Using the newly discovered unspecific peroxygenase *Tan*UPO from the ascomycetous fungus *Truncatella angustata*, 90% regioselective conversion of AA to 14,15-EET could be achieved. Selective conversion of AA to 18-HETE, 19-HETE as well as to 11,12-EET and 14,15-EET was also demonstrated with known peroxygenases, i.e., *Aae*UPO, *Cra*UPO, *Mro*UPO, *Mwe*UPO and *Cgl*UPO. The metabolites were confirmed by HPLC-ELSD, MS^1^ and MS^2^ spectrometry as well as by comparing their analytical data with authentic standards. Protein structure simulations of *Tan*UPO provided insights into its substrate access channel and give an explanation for the selective oxyfunctionalization of AA. The present study expands the scope of UPOs as they can now be used for selective syntheses of AA metabolites that serve as reference material for diagnostics, for structure-function elucidation as well as for therapeutic and pharmacological purposes.

## 1. Introduction

Lipid mediators are short-lived bioactive molecules which are rapidly formed after cell activation. Their physiological and pathophysiological functions are diverse and only partially understood. They are known to be involved in the functionality of the muscle and nervous systems [1,2,3,4], they regulate inflammatory processes [5,6], modulate immune responses [7,8,9,10], participate in the defense against allergies and parasites [11], and regulate vascular function [12,13] as well as platelet aggregation [14,15]. A well-known class of lipid mediators are the arachidonic acid-derived oxygenated eicosanoids.

Arachidonic acid [AA; (5Z,8Z,11Z,14Z)-5,8,11,14-eicosatetraenoic acid] is a long-chain polyunsaturated fatty acid (PUFA) belonging to the ω-6 fatty acids (FA). It consists of 20 carbon atoms with four double bonds in *cis*-configuration resulting in a hairpin structure. The compound also exhibits a degree of flexibility to interact with proteins and helps maintain the fluidity of cell membranes [16,17]. Physiologically, AA is present in the cytoplasmic membrane in the form of phospholipids. In response to an external signal, AA is released by, among others, phospholipase A2 (EC 3.1.1.4), which mediates hydrolysis of the phospholipid backbone, resulting in the free AA molecule [18]. Free AA is then metabolized mainly by three types of enzymes: cyclooxygenase (COX; EC 1.14.99.1), lipoxygenases (LOXs; EC 1.13.11.x), and cytochrome P450 monooxygenases (CYPs or P450s; EC 1.14.x.x) [19]. Via the COX pathway, AA is converted into prostaglandins (PGs) and thromboxanes (TXs) [20]. LOX enzymes can convert AA into leukotrienes (LTs) and lipoxines (LXs) [21,22,23,24]. In addition, epoxyeicosatrienoic acid (EETs) and hydroxyeicosatetraenoic acid (HETEs) can be formed via a P450 pathway (Figure 1) [25,26,27,28]. While the COX- and LOX-derived metabolites PGs, TXs, and LTs tend to be classified as pro-inflammatory, P450-derived eicosanoids partly possess anti-inflammatory and inflammation-resolving properties.

The protein superfamily of P450s in humans consist of 57 CYP isoenzymes, with subtypes (families) CYP1, CYP2 and CYP4 hydroxylating AA primarily by means of ω-oxidation [29]. Thereby, a hydroxyl group is attached to the terminal (ω, C20) and/or subterminal (ω-1 to ω-4, C16–C19) carbons. The molecular oxygen (O_2_) required for the reaction is reduced to a hydroxyl group (R-OH) and water (H_2_O), while at the same time the cosubstrate NAD(P)H is oxidized [30]. The terminal hydroxylated 20-HETE is formed by members of the CYP4A and CYP4F families. Unlike the other oxyfunctionalized AAs, 20-HETE has vasoconstrictive and pro-inflammatory effects and has been linked to cardiovascular diseases as angiotensin II-induced hypertrophy [31,32]. In contrast, the subterminal hydroxylated HETEs (16–19-HETE) are thought to have mainly anti-inflammatory effects [33]; of these, 19-HETE has received most attention to date. 19-HETE that is mainly formed by CYP2C19, CYP1A2, and CYP4A11, has a direct antagonistic effect by counteracting the vasoconstrictor effects of 20-HETE on the kidneys [34,35].

The members of the CYP2C and CYP2J families exhibit predominantly epoxygenase activity leading to different EETs. These emerge after epoxidation of one of the four *cis*-configured double bonds, forming 5,6-, 8,9-, 11,12-, and 14,15-EETs. In particular, the latter have anti-inflammatory properties and support the body’s own inflammation resolution [36,37,38]. Due to the promising potential of these CYP-based metabolites, their prospective value as therapeutic agents against diseases such as COVID19 has recently been reviewed [39]. Since EETs are rather short-lived and unstable due to rapid inactivation by soluble epoxide hydrolases [40,41], their availability poses major challenges to researchers. To study the effects of these anti-inflammatory eicosanoids in more detail, a simple and cheaper alternative to chemical synthesis would be needed. EETs are chemically difficult to prepare since the corresponding methods tend to be non-selective, require strongly acidic conditions (e.g., peracids) and necessitate multistep separation and purification stages, resulting in low overall yields (~10% as a mixture of all epoxides) [42,43,44]. Due to the demand for high purity, higher yields and fewer production steps, bio-based processes can help improve existing chemical procedures. A biological system that allows regio- and enantioselective epoxidation of polyunsaturated fatty acids is the P450 enzyme BM3 from *Bacillus megaterium*. Here, AA is regioselectively converted into 14,15-EET and obtained as an enantiomerically pure methyl ester with up to 52% yield (mg-scale) [45].

Another example of promising oxyfunctionalizing biocatalysts are the unspecific peroxygenases (UPOs, EC 1.11.2.1). As with P450s, UPOs are glycosylated heme-thiolate proteins, but belong to another protein superfamily and noteworthy, most are extracellular proteins secreted by fungi. Up to date, several thousand putative UPO sequences (>4000) have been found in fungal genomes [46]. In this context, they occur throughout the fungal kingdom, including most phyla of the true fungi (Eumycota) as well as some pseudo-fungal stramenopiles (Peronosporomycetes, formerly ‘Oomycota’) [47,48]. Phylogenetically, these enzymes can be classified into two large families: the ‘short’ UPOs (family I) and the ‘long’ UPOs (family II), which differ, among others, in molecular size [49]. Catalytically, UPOs resemble the human CYP enzymes (e.g., regarding substrate spectra as well as product patterns). Unlike CYPs (and most other P450s) working with O_2_ that must be activated, UPOs react with reactive hydroperoxides (R-OOH, typically H_2_O_2_) as oxygen source and electron acceptor. Thus, they do not require additional redox and/or electron-transfer partners (NADH, flavin, ferredoxin) for their function. In other words, UPOs combine the catalytic activities of heme peroxidases with the versatile oxygen-transfer potential of P450s/CYPs [47,50,51,52,53]. UPOs have already been used for a variety of reactions, including the epoxidation of selected unsaturated fatty acids, fatty acid methyl esters, and vegetable oils [54,55,56]. Therefore, UPOs are also promising candidates for the selective synthesis of various PUFA metabolites. In this study, the regioselective oxyfunctionalization of arachidonic acid to EETs and HETEs by UPOs is demonstrated.

## 2. Materials and Methods

### 2.1. Chemicals

Arachidonic acid (5Z,8Z,11Z,14Z-eicosa-5,8,11,14-tetraenic acid) was obtained from MP Biomedicals Germany GmbH (Eschwege, Germany). The following HETEs and EETs were ordered from Cayman Chemical (Ann Arbor, Michigan, USA): (±) 18-HETE ((±) 18-hydroxy-5Z,8Z,11Z,14Z-eicosatetraenoic acid), 19(S)-HETE (19S-hydroxy-5Z,8Z,11Z,14Z-eicosatetraenoic acid), 19(R)-HETE (19R-hydroxy-5Z,8Z,11Z,14Z-eicosatetraenoic acid), (±) 11(12)-EET ((±) 11,(12)-epoxy-5Z,8Z,14Z-eicosatrienoic acid) and (±) 14(15)-EET ((±) 14(15)-epoxy-5Z,8Z,11Z-eicosatrienoic acid). All other chemicals were obtained from VWR International GmbH (Darmstadt, Germany) if not indicated otherwise. The chemicals used had reagent-grade purity or were analytical standards.

### 2.2. Enzymes

The conversions of arachidonic acid were carried out with several wild-type UPOs (main isoforms) isolated from liquid cultures of basidiomycetous and ascomycetous fungi. The basidiomycetous fungus *Cyclocybe (Agrocybe) aegerita* (order Agaricales, ‘agarics’) secreting well-known *Aae*UPO (46 kDa) is deposited at the DSMZ (*Deutsche Stammsammlung für Mikroorganismen und Zellkulturen*; Braunschweig, Germany) under the accession number DSM 22459. Other agarics used for enzyme production were *Coprinellus radians* (DSM 888; *Cra*UPO, 45 kDa), *Marasmius rotula* (DSM 25031; *Mro*UPO, 32kDa) and *Marasmius wettsteinii* (DSM 106021; *Mwe*UPO 32 kDa). *Chaetomium globosum* (DSM 62110) is an ascomycetous fungus (mold) of the order Sordariales producing *Cgl*UPO (36 kDa). All of these enzymes were isolated and purified as described previously [57,58,59,60,61]. Novel *Tan*UPO described herein was prepared with an own isolate of the ascomycetous fungus *Truncatella angustata* (order Xylariales). The cultivation of *Truncatella angustata* was performed in a semisynthetic medium under the conditions summarized in Appendix A [62]. Cultivation process and physicochemical characteristics of purified *Tan*UPO are given in the Appendix A) [63,64]. Purification of the enzyme was achieved by multistep fast protein liquid chromatography (FPLC) using size exclusion, ion exchange and hydrophobic interaction techniques on appropriate separation columns (SEC, IEC, HIC) depending on the particular UPO (Appendix A). Purity of the enzymes was verified by UV/Vis spectroscopy and sodium dodecyl sulfate polyacrylamide gel electrophoresis (SDS-PAGE) (Appendix A).

Glucose oxidase (GOx, EC 1.1.3.4) from *Aspergillus niger* was purchased from Sigma-Aldrich (Taufkirchen, Germany). The specific activity given was 224,890 U g^−1^ where one unit (1 U) is equivalent to the oxidation of 1 µmol of *β*-D-glucose to D-gluconolactone and hydrogen peroxide per min at pH 5.1 and 35 °C.

### 2.3. Enzyme Assays

Activities of the above mentioned UPOs (except *Tan*UPO) were routinely measured photometrically by monitoring the oxidation of 5 mM veratryl alcohol (VA; 3,4-dimethoxybenzyl alcohol) to corresponding veratraldehyde (3,4-dimethoxybenzaldehyde) at 310 nm (ε_310_ = 9300 M^−1^ cm^−1^) in McIlvaine buffer at pH 7.0 for *Aae*UPO, *Cra*UPO and *Cgl*UPO, and at pH 5.5 for *Mro*UPO and *Mwe*UPO. Reactions were started by the addition of hydrogen peroxide (2 mM). One unit (1 U) corresponds to the oxidation of 1 μmol of VA to veratraldehyde within 1 min.

Activity of *Tan*UPO was first measured by the oxidation of ABTS [2,2′-azino-*di*(3-ethylbenzthiazoline-6-sulfonate) *di*-ammonium salt] to the corresponding cation radical (ABTS^+•^) at 420 nm (ε_420_ = 36,000 M^−1^ cm^−1^) in McIlvaine buffer at pH 4.5. Here, ABTS was used as substrate instead of VA since the catalytic efficiency of *Tan*UPO for ABTS is in the same order of magnitude as that for VA obtained by the other UPOs (Appendix A). The reaction was started by adding hydrogen peroxide (2 mM).

### 2.4. Enzymatic Conversion of Arachidonic Acid

To study the enzymatic conversion of arachidonic acid, we used 1-mL glass vials containing 500 µL reaction mixture with 2 U mL^−1^ of different UPOs in 20 mM phosphate buffer (pH 7.0). The substrate (AA) concentration was 1 mM, and 20% (*v*/*v*) acetone served as co-solvent. The co-substrate hydrogen peroxide (H_2_O_2_) was continuously generated by glucose (2%) and glucose oxidase (0.04 U mL^−1^) [65]. The vials were incubated on a rotary shaker at 30 °C and 800 rpm. After 2 h, the reaction was stopped by adding 500 µL cooled acetonitrile (−20 °C), followed by centrifugation at 17,000× *g* and 4 °C for 20 min. The supernatant was analyzed by HPLC and MS (see Section 2.5). For kinetic measurements, 50-µL samples were taken at different time points (between 0 and 180 min) and reactions were stopped immediately as described above.

### 2.5. Analytical Methods

#### 2.5.1. High Performance Liquid Chromatography/ELSD

The analytical studies of AA and its oxyfunctionalized metabolites was performed with an HPLC system LaChrom Elite^®^ (VWR-Hitachi, Radnor, PA, USA) consisting of the pump system L-2130, the autosampler L-2200, the column oven L-2300 and the diode array detector (DAD) L-2455 coupled with a low-temperature evaporative light scattering detector (ELSD Sedex 100, Sedere, Alfortville Cedex, France). The control of the system and the data evaluation were carried out by the EZChrome Elite^®^ software. A Kinetix^®^ reversed phase column (C18, 5 μm, 100 Å, 150 × 4.6 mm, Phenomenex, Torrance, CA, USA) with a corresponding pre-column was used for separation. Two mobile phases, A (diH_2_O, 0.1 % (*v*/*v*) formic acid) and B (acetonitrile, 0.1 % (*v*/*v*) formic acid), were used as eluents. The injection volume was 20 µL and the separation of analytes was performed at 40 °C and a flow rate of 1 mL min^−1^ using the following gradient: 0 min, 25% B; 2 min, 25% B; 12 min, 95% B; 17 min, 95% B; 17.1 min, 25% B; 20 min, 25% B. For detection of eluted PUFAs by ELSD, a drift temperature of 50 °C, 2 s laser filter, a measuring rate of 200 ms and nitrogen (N_2_) as carrier gas were used. All solvents were filtered and degassed prior to use. Analytes were identified using authentic standards and high-resolution mass spectrometry (HRMS).

#### 2.5.2. High Resolution Mass Spectrometry

Additional analytical investigation of AA and its metabolites was performed on a Thermo Scientific Vanquish Flex Quaternary UHPLC system (Thermo Fisher Scientific, Waltham, MA, USA) using a Kinetex^®^ EVO column (C18, 5 µm, 100 Å, 150 × 4.6 mm, Phenomenex). The injection volume was 1 µL and the column was eluted at a flow rate of 0.5 mL min^−1^ and 40 °C with two mobile phases, A (diH_2_O, 0.1% formic acid) and B (acetonitrile, 0.1% formic acid), applying the following gradient: 0 min 25% B; 2 min, 25% B; 12 min, 95% B; 17 min, 95% B; 17.1 min, 25% B; 20 min, 25% B.

MS^1^ and MS^2^ spectra were obtained using a Thermo Scientific Q Exactive Focus quadrupole-Orbitrap mass spectrometer (Thermo Electron, Waltham, MA, USA) coupled with a heated electrospray ionization source in the negative mode. The tune operating parameters were as follows: the rate of sheath gas flow and auxiliary gas flow were 40 and 15 (arbitrary units), respectively; spray voltage 4.0 kV; the temperature of capillary and auxiliary gas heater were 260 °C and 400 °C, respectively; high-resolution MS was operated at full scan mode (MS^1^) with a mass range of *m*/*z* 100–1500 at a resolution of 70,000 (*m*/*z* 200). The MS^2^ data at a resolution of 35,000 were obtained by parallel reaction monitoring mode triggered by inclusion ions list, which was built by the molecule predicted. The collision energy varied between CE15 for epoxides and CE20 for hydroxylated metabolites.

## 3. Results

### 3.1. Small-Scale Conversion of AA by Different UPOs

To investigate the general possibility of oxyfunctionalization of AA with UPOs, a screening was first carried out with different UPOs at 30 °C and pH 7.0 for 2 h. A short enzyme cascade with glucose oxidase and glucose was used to provide the oxidant hydrogen peroxide. Reactions were run in the presence of acetone (20% *v*/*v*) to increase the solubility of AA. Six homologously prepared UPOs (i.e., non-recombinant wild-types) were tested: *Tan*UPO, *Cgl*UPO, *Mro*UPO, *Mwe*UPO, *Aae*UPO and *Cra*UPO (see entries 1–6 in Table 1). The optimal reaction conditions (pH, temperature) were previously determined in terms of best product yield by new *Tan*UPO (Appendix A).

Analyses with HPLC-ELSD and HPLC-MS revealed four main products of AA conversion. While the well-known long UPOs (*Aae*UPO and *Cra*UPO) formed the hydroxylated metabolites 18- and 19-HETE, the short UPOs (*Mro*UPO, *Mwe*UPO, *Cgl*UPO) as well as the novel *Tan*UPO preferred the epoxidation of AA into EETs (Table 1). Each of the tested UPOs was able to convert AA, with *Tan*UPO and *Cra*UPO showing the highest and lowest conversion rates of 95% and 70%, respectively. In the MS^1^ spectra, each of the four main metabolites exhibited an abundant [M − H]^−^ ion at *m*/*z* 319 indicating the insertion of one oxygen atom (*m*/*z* + 16) (Appendix A). The four metabolites could be distinguished and identified on the basis of their retention times both in HPLC-ELSD and HPLC-MS elution profiles, as well as on the basis of their fragmentation patterns in the MS^2^ spectra and in comparison with appropriate reference standards (Figure 2 and Appendix A, Table 2).

Noteworthy, *Tan*UPO converted about 90% of AA exclusively into 14,15-EET. The retention time (RT) of the product detected by HPLC-ELSD was 13.0 min (Figure 2A, III). The associated product ions [M − H]^−^ in the MS^2^ spectra were *m*/*z* 219 (loss of the neutral aldehyde with proton rearrangement), *m*/*z* 175 (loss of CO_2_ from the previously formed fragment) and *m*/*z* 113 (an enolate anion fragment) (Table 2, III; Appendix A). Analysis of the time course of AA oxidation (Figure 3A) revealed that the formation of the product 14,15-EET reached its maximum between 90 and 120 min (calculated by the relative peak area of the corresponding peak to the total areas detected by ELSD in %), and then decreased again. With increasing time, another product appeared (Figure 2A, III) with a RT of 10.7 min and an associated [M − H]^−^ ion of *m*/*z* 335. This indicates the incorporation of two oxygen atoms into the AA molecule. The respective product ions indicate further hydroxylation (Appendix A, VII A).

The conversion of AA by *Cgl*UPO led also to the formation of 14,15-EET (Figure 2A, III) with the same fragment ions as described above, but also to the formation of 11,12-EET. The RT of the latter metabolite (HPLC-ELSD) was 13.3 min (Figure 2A, II) and the related product ions [M − H]^−^ in the MS^2^ spectra were *m*/*z* 167, *m*/*z* 208 and *m*/*z* 179 (Appendix A; Table 2, II). The products were obtained in a ratio of 4:1 (14,15-EET:11,12-EET) after 120 min. The amount of both products decreased again as the reaction progressed (120 to 180 min), which was accompanied by the formation of another product (RT 11.6 min, Figure 3B). According to the MS^1^ analysis, an [M − H]^−^ ion of *m*/*z* 335 was formed, which again indicates the incorporation of two oxygen atoms. However, the RT of the product was 11.7 min, which is different from that of the *Tan*UPO metabolite (Figure 2A, VII B). The MS^2^ fragmentation gave the following product ions: *m*/*z* 179, *m*/*z* 167 and *m*/*z* 113 indicating the formation of a double-epoxidized metabolite at positions C11, C12, C14 and C15 (Appendix A, VII B).

The two short UPOs, *Mro*UPO and *Mwe*UPO, showed similar product patterns to *Cgl*UPO, with 14,15-EET and 11,12-EET being the major products formed at ratios of 1.7:1 and 3.6:1, respectively (Figure 2A, II & III). The fragment ions confirming the products are listed in Table 2, II and III. Time courses of the reactions are shown in Figure 3C, D, and as in the case of the other UPOs, the amount of products decreased again in the period from 120 to 180 min, with some smaller product peaks appearing in the RT range from 10 to 12 min (Figure 2A, VII A). Of these peaks, the strongest signal was obtained at an RT of 10.7 min, indicating the same by-product formation as with *Tan*UPO. The MS^1^ (*m*/*z* 335) and MS^2^ spectra supported this assumption (Appendix A, VII A). The possible di-epoxide at an RT of 11.7 min (as with *Cgl*UPO) was apparently also formed, but in much smaller amount.

The long UPOs, *Aae*UPO and *Cra*UPO, showed a completely different product pattern with the two hydroxylated metabolites, 18-HETE and 19-HETE, as major products in the ratio 1:1.4 and 1:1.5, respectively. In the HPLC-ELSD profiles (Figure 2B, IV), 18-HETE appeared at RT 11.4 min and the associated product ions in the MS^2^ spectra were *m*/*z* 261 and *m*/*z* 217 (loss of CO_2_ from a previously formed fragment) (Appendix A; Table 2, IV). For 19-HETE, the RT of 11.8 min and the related product ions were *m*/*z* 275 and *m*/*z* 231 (Figure 2B, V and Appendix A; Table 2, V). While *Cra*UPO mainly formed these two metabolites, *Aae*UPO formed yet another product that appeared as a shoulder immediately adjacent to the 18-HETE peak in the chromatogram (Figure 2B, VI). According to the MS^1^ data, an ion with *m*/*z* 317 was present there. This implies the formation of an oxo-functionality (ketone) at the site of the hydroxyl group due to an over-oxidation activity (via a *gem*-diol intermediate). MS^2^ analysis supports these assumptions (Appendix A, VI). Although the mass signals of corresponding ketones were found at both C18 and C19, the C19-ketone (19-oxo-ETE) was clearly more present, which is why it is the only one shown (Figure 2, IV).

### 3.2. TanUPO Protein-Structure Simulation and Ligand Docking

Of the UPOs studied here, only the crystal structures of *Aae*UPO and *Mro*UPO exist to date. Therefore, protein models were generated to obtain information about the structure of the substrate channel of the new *Tan*UPO. Peptide mapping was used to assign the isolated protein of *Tan*UPO to a sequence in the genome of *Truncatella angustata* (GenBank: KAH8203164.1) (Appendix A). The sequence obtained was used for protein structure simulations (method description in Appendix A) and compared with *Aae*UPO and *Mro*UPO in terms of ligand dockings (Figure 4).

Five models of N-terminally processed (starting with amino acid 22 of NCBI # KAH8203164.1) *Tan*UPO were predicted by ColabFold and ranked (from 1 = “most confident” to 5) according to the AlphaFold2 prediction confidence-measures pLDDT (predicted local distance different test) (see Appendix A) and PAE (predicted alignment error) (see Appendix A). All models are quite similar on backbone position basis (Appendix A), with pLDDT values generally above 90 (high confidence), with a few exceptions in the 80 s range (good confidence). Local minima (troughs) of the pLLDT plot per-position are generally located in short loops (e.g., around amino acid 86, 105, 151, or 172) that connect helices with limited overspill to adjacent helix-residues (see Appendix A). Only the backbone position of the first three N-terminal amino acids and the last approximately 15 C-terminal amino acids exhibit pLDDT values <50. These parts of the models should not be part of later conclusions. This is in accordance with the AlphaFold protein structure database FAQs (https://alphafold.ebi.ac.uk/faq#faq-5 (accessed on 6 January 2022)), which provide guidelines for the interpretation of AlphaFold2 confidence measures. The 15 C-terminal amino acids were predicted as unstructured ribbon, and while most regions within a predicted model occupy well defined relative positions (low PAE; see Appendix A), this region exhibits high PAE in all models, consistent with low MSA (multiple sequence alignment) coverage (see Appendix A) in this region.

Since there was no appreciable difference between the predicted models in the positioning of amino acids that directly form the substrate access channel (SAC) of *Tan*UPO (in a CEAlign alignment of all five models; not shown), the 5th ranked model was chosen out of convenience as input to FlexAid for docking AA to the SAC and for other visualizations, as there the C-terminal ribbon “tail” obstructed the line of sight into the SAC the least (see Appendix A).

The SACs of *Tan*UPO, *Aae*UPO and *Mro*UPO are formed largely by structurally homologous helices (Ha = cyan, Hb = pink and Hc = green in Figure 4 and Appendix A) that run diagonally to the heme plane to varying degrees. In comparison, the side-chain residues of Ha and Hb in the three UPOs delimit the opposite sides of the SAC and most of the bottleneck, as well as largely the heme-cavity proximal to the channel. Hc runs approximately perpendicular to Ha and Hb and is mainly positioned “above” towards the outer protein surface. Thus, it blocks one side of the “upper” SAC between Ha and Hb, while the “lower” bottleneck part of the SAC on this side is either blocked by bulky phenylalanine residues from Ha and/or Hb (e.g., *Aae*UPO: F66 and F196, *Mro*UPO: F160) or it opens into a separate pocket above and offset to the heme (there are no phenylalanine residues on this flank in *Tan*UPO). The SAC side opposite to Hc is either bounded by side-chain residues of Hb and/or Ha (e.g., *Aae*UPO: F73 F188, T189, *Tan*UPO: L153, F157) or, in the case of *Aae*UPO and *Mro*UPO (at the bottleneck), involving a side-chain residue from a loop (*Aae*UPO: F118, *Mro*UPO: I84; La = sand color in Figure 4).

The SACs of *Aae*UPO and *Tan*UPO are slightly elliptic in cross-section and similar in shape to, e.g., bell-shaped flowers of *Digitalis purpurea* (foxglove) tapering slightly downward to a bottleneck into the heme cavity, whereas *Mro*UPOs has approximately the shape of an eccentric elliptical paraboloid with minimum radius comparable to the average radius of the channel of *Aae*UPO at any point along the channel towards the bottleneck.

FlexAid found plausible (in light of the identified product-spectra) fits of AA into each UPO enzyme and a single solution was selected each, to illustrate differences in the SAC morphology in order to add structural insight to the observed product-spectra of the compared UPOs.

## 4. Discussion

Our results demonstrate that UPOs can be used for the synthesis of active lipid mediators such as EETs and subterminal hydroxylated HETEs. In particular, EETs have interesting anti-inflammatory properties and support endogenous inflammation resolution [66,67,68]. Although their mechanisms of action and function have not been fully explored, they are already being discussed as potential therapeutic agents or therapeutic targets [69,70]. In terms of a possible therapeutic target, the increase of physiological EET levels by the use of soluble epoxide hydrolase inhibitors (sEHI) should be considered. In this regard, sEHI could reduce rapid degradation of EETs and thus prolong their biochemical presence (bioavailability) [71]. External administration of EETs would also be a way to increase and stabilize EET levels. This is countered by the fact that the availability of EETs as chemicals is low and they are therefore very expensive, as their synthesis is difficult to realize by chemical means.

In conventional chemical synthesis, epoxidation of PUFAs can be achieved with peracids such as *meta*-chloroperoxybenzoic acid (*m*CPBA). For this purpose, arachidonic acid is methylated with diazomethane in the first step. Subsequent extraction is followed by epoxidation of the *cis*-double bonds to *cis*-epoxides with *m*CPBA. Since this method is neither regio- nor enantioselective, complex separation and purification steps, such as chiral chromatography, are required to obtain pure compounds. The yield of all epoxides in a mixture is only 10% (mg scale). Moreover, these preparations are complicated by limited solubility, instability of reactants and products, and the associated challenging storage conditions [42,43,72].

The regio- and stereoselective preparation of HETEs is also challenging. Total synthesis of such lipid mediators has been reported in the literature, but the methods are time-consuming and require multiple steps. For example, the synthesis of 20-HETE requires seven steps. Starting from methyl 5-hexynoate, two crucial steps are necessary: one is the Cu-mediated formation of a C-C bond and the other is partial alkyne hydrogenation. The yield of the target molecule 20-HETE is less than 1% (mg scale) [73].

To realize a simpler, lower-cost alternative to chemical synthesis, the synthesis steps, the use of chemicals, and the purification steps would have to be reduced. Such a system could be realized with the help of biocatalysis. In the literature, conversion of arachidonic acid has already been reported (e.g., with the P450 BM3 from *Bacillus megaterium*) [45]. This bacterial P450 enzyme has the advantage of being a natural fusion protein from the P450 domain and an NADPH-dependent cytochrome P450 reductase [74]. Thus, no further electron transport protein is required. This approach represents a simpler and more cost-effective option compared to chemical synthesis of EETs but still suffers from the use of costly NADPH as an electron donating co-substrate.

UPOs have similar catalytic properties to P450/CYP enzymes and therefore allow convenient studies of human-like metabolite formation. They have the advantage of being better water-soluble and require only hydrogen peroxide for activation and function (i.e., no expensive cofactors or electron transport proteins are needed). They are furthermore particularly promising for the scale-up of batch processes. Among the UPOs tested here, the newly isolated *Tan*UPO proved to be particularly promising. In addition to the well-known short UPOs, *Mro*UPO, *Mwe*UPO, and *Cgl*UPO, which have already been extensively studied in terms of their versatile catalytic properties, such as epoxidation of lipids and fatty acids [54,55,56,61,75], *Tan*UPO showed here the ability to highly selective epoxidize an ω-6 fatty acid. The enzyme converted more than 90% of AA into a single metabolite, 14,15-EET.

Regarding the catalytic performance of AA conversion (into 14,15-EET), *Tan*UPO showed a *K*_m_ value of ~200 µM and a turnover number (*k*_cat_) of 1.6 s^−1^ (Appendix A). In comparison, P450s (CYP epoxygenases) in rat heart microsomes were reported to have a higher *K*_m_ value of about 570 µM for the same substrate [76]. The heart is the organ with the highest levels of 14,15-EET. Nevertheless, the affinity of these P450s toward the substrate AA is at least only half that of *Tan*UPO. However, microsomes not only contain CYP epoxygenases forming 14,15-EET but many other membrane-bound enzymes, so that a direct comparison of catalytic properties seems to be of little use here. When comparing the catalytic constants of *Tan*UPO with a heterologously expressed CYP2C8 enzyme, the *K*_m_ is about ten times higher (200 µM and 19.4 µM, respectively) [77]. On the other hand, the turnover number (*k*_cat_) has also to be considered, which is more than 1300 times higher for *Tan*UPO. Thus, the catalytic efficiency (k_cat_/*K*_m_) for the conversion of AA into 14,15-EET by *Tan*UPO (7.9 × 10^3^ M^−1^s^−1^) is about 130 times higher than that of CYP2C8 (59 M^−1^ s^−1^). In the study cited here [77], directed evolution was applied and two mutants of the CYP2C8 wild-type (WT) were expressed. As the result, the turn over number could be increased up to 160-fold, increasing the catalytic efficiency to 4.1 × 10^3^ M^−1^ s^−1^, which is still somewhat lower than that of *Tan*UPO. The CYP2C8 WT and mutants as well as the rat microsmes exhibited Michaelis Menten kinetics. In contrast, *Tan*UPO showed a sigmoidal kinetic (Appendix A). This effect can be explained by the buffer concentration. While here a relatively low-concentrated phosphate buffer (20 mM) was used, in the aforementioned studies a five-times higher concentration (100 mM) was applied. Similar effects were already observed with P450 BM3 using palmitate in varying phosphate buffer concentrations (10 to 100 mM) [78]. It was found that the kinetics at higher buffer concentrations (>50 mM), tended to follow the Michealis-Menten plot, while at lower concentrations (<50 mM) the Hill equation was a better fit (sigmoidal curve shape). This effect was attributed to homotropic allosteric interactions at low substrate concentrations [79]. The catalytic properties also include the product range. *Tan*UPO forms only one EET isomer whereas CYP epoxygenases such as CYP2C8 usually form mixtures of 11,12-EET and 14,15-EET [77,80,81]. Using the aforementioned CYP BM3 fusion protein, regioselective synthesis to form 14,15-EET was demonstrated as well (but with only 52% yield of the corresponding methyl ester) [45].

Besides *Tan*UPO, the other short UPOs and the long UPOs (*Aae*UPO, *Cra*UPO) formed either yet another EET isomer or 18-/19-HETEs. These findings, especially with respect to *Tan*UPO, are supported by corresponding protein-structure simulations [75,82,83,84,85,86]. In Figure 4, the UPO-heme is buried in a cavity within the protein that is connected to the protein surface by a SAC, running roughly perpendicular to the heme-plane. The SAC-topologies of *Aae*UPO (Figure 4CI,CII), *Mro*UPO (Figure 4BI,BII) and *Tan*UPO (Figure 4AI,AII) are very similar (see Appendix A), as indicated by the use of a common color-scheme for structurally homologous helices and short loops in Figure 4 and Appendix A. Helices Ha and Hb and the short loop La are nearly identical in length and relative position in all three UPOs. The protein domain forming helix Hc has fewer turns towards the N-terminus in *Aae*UPO than in the short UPOs, and the backbone of *Aae*UPO residues 237–239 more closely resembles a ribbon. The protein domains that extend the SAC of *Aae*UPO toward the protein surface (shown in yellow and blue in Figure 4, Appendix A, respectively) do not have homologous domains in the short UPOs, *Mro*UPO and *Tan*UPO.

As the term “unspecific” implies, individual UPO enzymes can have a broad substrate spectrum and form a range of products depending on the substrate used [48]. In this context, AA is a particularly interesting molecule (as the substrate of different UPOs) since it has a fairly long and flexible carbon chain and is peroxygenated at different positions (sometimes by the same enzyme) along its carbon-chain (Table 1). Using the experimentally determined product spectra of *Aae*UPO, *Mro*UPO, and *Tan*UPO, Figure 4 shows only some examples of plausible FlexAID-generated fits of AA into the respective SACs. All SACs are quite hydrophobic and differ mainly in the composition of hydrophobic aliphatic and hydrophobic aromatic residues (see Appendix A; SAC residue #F/#A/#I/#L/#M: *Aae*UPO = 5/4/1/0/0, *Mro*UPO = 1/1/6/4/0, *Tan*UPO = 3/2/4/3/1).

*Mro*UPO has the widest (in one dimension) SAC of the three structurally compared UPOs, and AA can fit almost length-wise (shaped like the Latin letter “U”) into the *Mro*UPO SAC so that the C11,12 or C14,15 positions can face the heme. AA’s carbon-chain could interact primarily with leucine and isoleucine residues along the bow of the SAC, which would allow AA to “slide along” to some extent. This would be the minimum requirement for the formation of 11,12-EET and 14,15-EET. Positions C4,5 and C8,9 apparently cannot be sufficiently exposed to the heme, likely due to their relative proximity to AA’s charged carboxyl group (which is present as a carboxylate at pH 7.0) and the lack of appropriate side-chain residues to positively interact with this group within the *Mro*UPO SAC. In contrast, *Tan*UPO also forms 14,15-EET (but not 11,12-EET). Its SAC is much less eccentric than that of *Mro*UPO, and AA may need to thread into the SAC in a single pass. Oddly, *Tan*UPO is predicted to have a small extra pocket within its SAC, located to the side and above the heme, which could accommodate positions C17 to C20 of AA, making C14,15 accessible for oxygenation, but not C11,12. Additionally, R217 of *Tan*UPO could stabilize the position of the carboxylate of AA in such a fit.

*Aae*UPO has the longest SAC-interface that AA can potentially interact with. Similar to *Tan*UPO, AA would have to thread-through to get to the heme. AA is only hydroxylated at C18 or C19 but not further “up” the chain. The lack of EET products is plausible in that the *Aae*UPO SAC is quite narrow and AA has no spatial freedom to bend into it in a way to present anything but the most extreme ends of the molecule to the buried heme. Interestingly, a T-shaped π-stack runs (possibly coincidentally) along the bottleneck of *Aae*UPO between F66 and F196 (see Figure 4CI,CII), which is unique among the three UPOs.

In addition to the formation of the two HETEs, *Aae*UPO caused some over-oxidation leading to ketone (19-oxo-ETE) formation. Although the *Cra*UPO showed the lowest conversion for AA (70% within 2 h), it hardly caused such over-oxidation. Therefore, after optimization of the reaction, this enzyme would be the most promising candidate to prepare 18- and 19-HETE at larger scale. Nevertheless, a separation of the two HETEs would be necessary here as well. Alternative solutions, such as protein engineering via directed evolution, would be an elegant and promising approach of future studies.

Apart from the synthesis of known human HETEs and EETs from AA, the formation of a new AA metabolite was also demonstrated here. Thus, in addition to 11,12-EET and 14,15-EET, *Cgl*UPO catalyzed the formation of a product with two additional oxygen atoms (*m*/*z* 335). Based on the MS^2^ fragmentation, we suspect that it is a di-epoxidized product (Appendix A), which is not part of the human metabolite portfolio. The ability of *Cgl*UPO to form di-epoxides from vegetable oils has already been demonstrated [56]. In preliminary experiments, direct dosage of hydrogen peroxide was tested instead of continuous generation by glucose oxidase, resulting in a shift of the product pattern towards di-epoxides. In future experiments, it would therefore be interesting to isolate this new metabolite and test its biological efficacy.

## 5. Conclusions

The results presented here open up interesting possibilities in the field of biocatalysis with fungal unspecific peroxygenases, particularly with regard to the preparation of human AA metabolites usually formed by P450s/CYPs. This allows for easier access to these compounds (e.g., for diagnostic purposes or as reference materials). Since the field of CYP-based eicosanoids is still poorly understood compared to other lipid mediators (e.g., prostaglandins, leukotrienes), the metabolites produced by UPOs could be used to further elucidate the function and mode of action of these EETs and HETEs in cell culture experiments. Furthermore, these metabolites have promising application potential for therapeutic and pharmacological purposes.

## Figures and Tables

**Figure 1 microorganisms-10-01267-f001:**
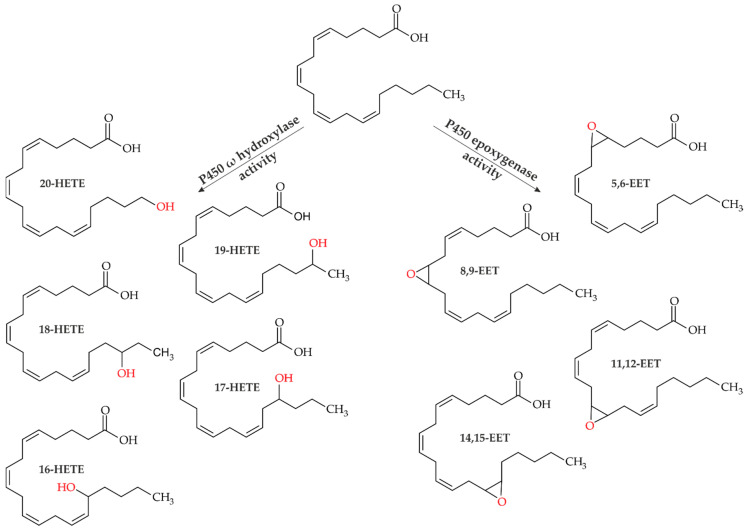
Arachidonic acid metabolism by cytochrome P450 monooxygenases. Left site: Hydroxyeicosatetraenoic acids (HETEs) 16-, 17-, 18-, 19-, 20-HETE formed by P450 ω-hydroxylase activity. Right site: Epoxyeicosatrienoic acids (EETs) 5,6-EET, 8,9-EET, 11,12-EET and 14,15-EET formed by P450 epoxygenase activity.

**Figure 2 microorganisms-10-01267-f002:**
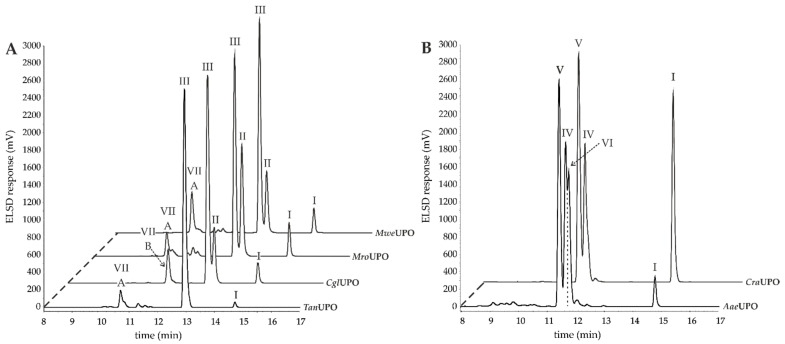
HPLC analysis (elution profiles recorded by ELSD) of products formed by different UPOs during the conversion of AA over 2 h. The reactions were carried out with 1 mM AA in 20 mM sodium phosphate buffer (pH 7.0), in presence of 20% (*v*/*v*) acetone, 2% (*w*/*v*) glucose, 2 U mL^−1^ UPO and 0.04 U mL^−1^ glucose oxidase. (**A**): HPLC-ELSD elution profiles of EETs formed. I: Arachidonic acid; II: 11,12-EET; III: 14,15-EET; VII A: Metabolite with *m*/*z* 335; VII B: metabolite with *m*/*z* 335. (**B**): HPLC-ELSD elution profiles of formed HETEs. I: Arachidonic acid; IV: 18-HETE; V: 19-HETE; VI 19-oxo-ETE.

**Figure 3 microorganisms-10-01267-f003:**
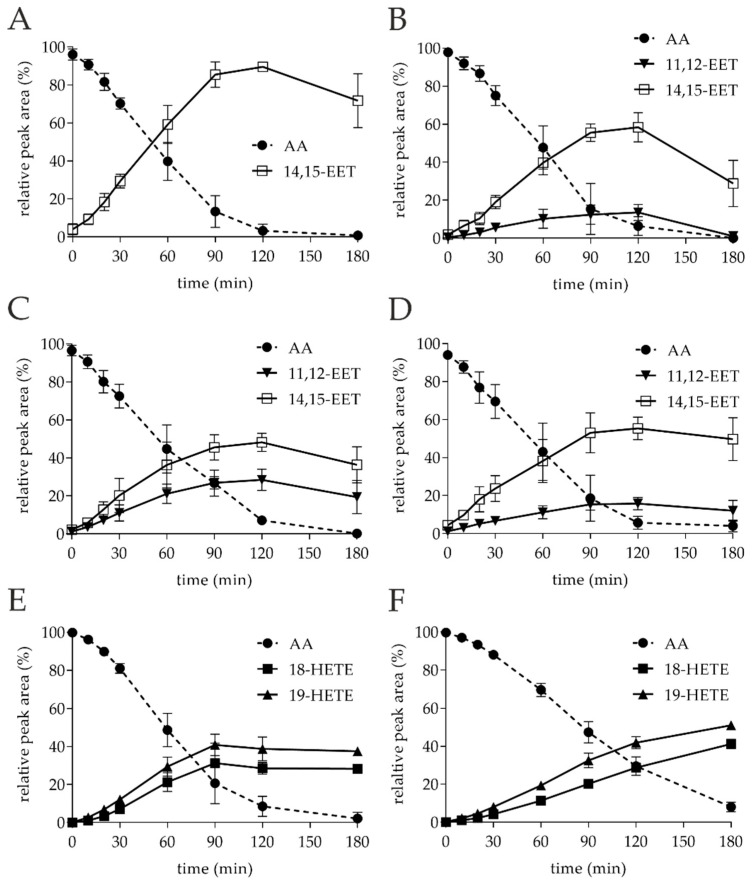
Time courses of AA conversion by different UPOs (0–180 min, *n* = 6). Reactions were carried out with 1 mM AA in 20 mM sodium phosphate buffer (pH 7.0) in the presence of 20% (*v*/*v*) acetone, 2% (*w*/*v*) glucose, 2 U mL^−1^ UPO and 0.04 U mL^−1^ glucose oxidase. The peak areas detected by HPLC-ELSD are given as relative peak areas (%) related to the total areas in the chromatograms. (**A**): *Tan*UPO; (**B**): *Cgl*UPO; (**C**): *Mro*UPO; (**D**): *Mwe*UPO; (**E**): *Aae*UPO; (**F**): *Cra*UPO.

**Figure 4 microorganisms-10-01267-f004:**
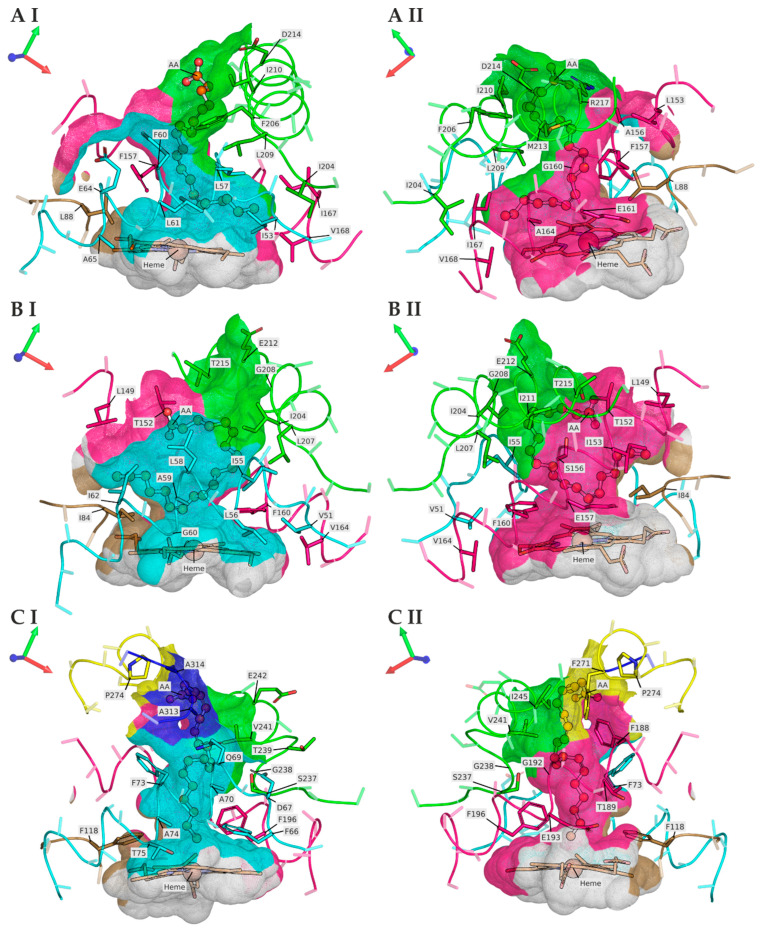
Carved solvent excluded surface (SES) of the heme-pocket and substrate access channel (SAC) of *Tan*UPO ((**AI**) + (**AII**)) (AlphaFold2 simulation), *Mro*UPO ((**BI**) + (**BII**)) (PDB#: 5FUJ Chain A) and *Aae*UPO ((**CI**) + (**CII**)) (PDB#: 2YOR chain A); orthoscopic view. Hydrogens are visually hidden. Protein backbone segments that outline the SAC are depicted as colored ribbons and the respective SES segments are colored identically (cyan, hot pink, sand, green, yellow, blue). The prosthetic heme and protein side-chains are depicted as sticks, the ligand arachidonic acid (AA) as ball-and-sticks and the heme-iron as a sphere. Protein side-chain residues are generally transparent and shown only for Cα to Cβ (with the exception of glycine) to reduce visual clutter. Side-chain residues of interest, on the other hand, are shown in full length and opaque. Black outlines were added to improve the visibility of obscured molecules and residues of interest. Labels are in the IUPAC one-amino-acid code. Stick-carbons are colored in either the color of the respective main-chain segment or in orange (heme and AA), oxygen atoms in red, nitrogen atoms in blue. All UPOs were structurally aligned using the CEAlign method (PyMol v. 2.2.5), and the coordinate axes in the upper left corner indicate the position (point-of-view) in the common 3D space.

**Table 1 microorganisms-10-01267-t001:** Screening of different UPOs for the oxyfunctionalization of AA ^1^.

Entry	Enzyme	AA Conversion Rate (%) ^2^	Yield 18-HETE (%) ^2^	Yield 19-HETE (%) ^2^	Yield 11,12-EET (%) ^2^	Yield 14,15-EET (%) ^2^
1	*Tan*UPO	95.6 ± 4.4	-	-	-	89.2 ± 4.8
2	*Cgl*UPO	92.8 ± 6.3	-	-	14.5 ± 3.1	58.0 ± 7.0
3	*Mro*UPO	91.4 ± 4.3	-	-	28.6 ± 4.7	48.1 ± 4.2
4	*Mwe*UPO	92.5 ± 6.3	-	-	15.6 ± 4.1	56.3 ± 5.5
5	*Aae*UPO	89.4 ± 6.2	27.0 ± 1.6	38.8 ± 4.9	-	-
6	*Cra*UPO	70.8 ± 5.7	28.4 ± 2.6	42.3 ± 3.6	-	-

^1^ Reactions were carried out at 30 °C for 2 h with 1 mM (304 µg mL^−1^) AA in the presence of 20% (*v*/*v*) acetone in 20 mM sodium phosphate buffer (pH 7.0) supplemented with 2% (*w*/*v*) glucose, 2 U mL^−1^ UPO and 0.04 U mL^−1^ glucose oxidase. ^2^ Relative peak area determined by HPLC-ELSD.

**Table 2 microorganisms-10-01267-t002:** MS^1^ and MS^2^ signals (in negative ionization mode) of the four major metabolites of AA formed by UPOs.

Peak	MS^1^ [M − H]^−^	MS^2^ Fragmentation [M − H]^−^ (Relative Abundance %)	Structure
II	319.2262 theor. for C_20_H_32_O_3_	167.1062 (100), 179.1063 (26), 59.0123 (CH_3_COOH) (24), 257.2265 (-H_2_O-CO_2_) (17), 301.2163 (-H_2_O) (15), 208.1089 (8)	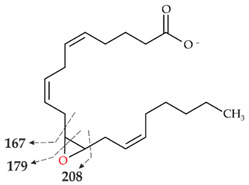
III	319.2261 theor. for C_20_H_32_O_3_	113.0955 (63), 257.2264 (-H_2_O-CO_2_) (51), 301.2159 (-H_2_O) (49), 59.0123 (CH_3_COOH) (45), 175.1475 (38), 219.1375 (35), 99.0798 (14)	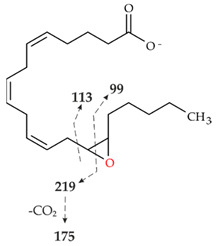
IV	319.2261 theor. for C_20_H_32_O_3_	261.1848 (86), 301.2159 (-H_2_O)(54), 59.0123 (CH_3_COOH) (49), 217.1947 (25), 257.2264 (-H_2_O-CO_2_)(15)	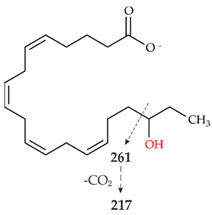
V	319.2261 theor. for C_20_H_32_O_3_	275.2003 (99), 59.0123 (CH_3_COOH)(74), 301.2159 (-H_2_O)(50), 231.2104 (20), 257.2264 (-H_2_O-CO_2_)(13)	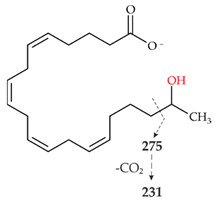

## Data Availability

The data presented in this study are available on request from the corresponding author.

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
