# Peer review of "Novel Unspecific Peroxygenase from Truncatella angustata Catalyzes the Synthesis of Bioactive Lipid Mediators"

_microorganisms, 2022, doi:10.3390/microorganisms10071267_

Round 1
Reviewer 1 Report
In the paper entitled “Novel unspecific peroxygenase from Truncatella angustata catalyzes the synthesis of bioactive lipid mediators”, Konig and coworkers described the capacity of a newly discovered unspecific peroxygenase TanUPO to convert arachidonic acid (AA) into epoxidized eicosanoids (EETs). They evaluate this activity in vitro and compare it to other unspecific peroxygenases, for which they showed their ability to convert AA into EETs or hydroxylated eicosanoids (HETEs). The last part of the paper is a modeling of the three-dimensional structure of TanUPO with the substrate and a comparison with the structures of two other UPOs, MroUPO and AaeUPO, in an attempt to explain the differences in product synthesis.
This article is interesting and contains a lot of experimental data. The authors should be congratulated for the large amount of work and their clear presentation. However, the article needs additional explanations and results before its publication.
Major comments:
1) Regarding the enzyme assays (page 4, § 2.3): why all the enzymes were not tested on ABTS? Only TauUPO was tested on ABTS and all the other on veratryl alcohol. It is important to make the determination of the enzyme unit on the same substrate because of their use to evaluate their ability to convert AA. Especially, because the TauUPO is described as the most efficient enzyme to produce 14,15-EET and in its overall ability to convert AA.
2) Kinetic parameters of TauUPO towards AA and optimal pH and temperature should be determined. As the authors described the first oxyfunctionalyzation of AA by TauUPO, it is necessary to add in this article a full characterization of the enzymatic behavior of TauUPO towards its substrate, as well as to determine its optimal reaction conditions.
3) The discussion should be focused on the enzymatic reactions rather than on the models of structure only. The discussion repeated the interest to obtain new enzyme to catalyze lipid oxidation, and especially noted that alternative to cytochrome P450 reductase, which use the costly NADPH, is required. It is very interesting but, the results obtained here were not discussed regarding the comparison of the efficiency of the two types (CYPs vs UPO) of enzyme to convert AA. Please, compare it to what is known from the literature. Moreover, too much space is dedicated to the structural aspects, which are only based on models for the TauUPO protein and the placement of the substrate in the three structures.
Minor comments:
1) Supplementary material: what was the total volume of culture used to prepare the TanUPO? How many flasks of 500-ml?
2) Figure S4 (line 191 supp data): SDS-PAGE, the names of the UPO are not correct, all ‘BacUPO’ whereas they should be those studied in Table 1.
3) The numbering of supplementary figures is wrong (S10 or S11?). It seams more appropriated to put them in the order of appearance in the text. The full MS spectra should appear at the beginning (they correspond to the figure 2).
4) Please, provide the correct accession number: the gene “g2685.t1” (supplementary data page 4, line 160) cannot be found in Mycocosm and the genbank accession JAJJMK000000000.1 refers to the whole genome assembly, and does not contain the protein sequence.
5) Some italics for the enzyme names are missing (legend of fig 4 for instance).
6) The sentence starting at line 483 (conclusion) is unclear, please rephrase it.
Author Response
Dear reviewer,
please see the attachement.
kind regards,
Rosalie König

Reviewer 2 Report
Suggestions to the Authors;
In this work, the authors carried out an extensive and very complete work where they evaluated the relevance of named unspecific peroxygenases in the selective syntheses of metabolites with application in therapeutic field, for example. Moreover, the results have been adequately justified and discussed based on previous publications and has a wide number of tables and figures where are summarized all data. In my opinion this work may be interesting as a point of support in future research, so I only comment to the authors on some suggestions to clarify some doubts that arose during this review.
1. In figure S1, why the pH value increase since 4.5 until 8 from about the tenth day? Could the authors give some explanation in the manuscript? Was relevant this fact in results obtained?
2. In section “3.1. Small-scale conversion of AA by different UPOs” below table 1 appear “2Determined by HPLC-ELSD”. In my opinion this comment is moved and the correct place is not the indicate in the text.
3. The formal form to indicating the Celsius degrees is writing number-spacing-°C (e.g. 37 °C). Revise the whole text and correcting, in some parts of manuscript is correct but other is incorrect.
Author Response

(The authors gave the same response as above.)

Reviewer 3 Report
Comments to AuthorIn this manuscript, the author using the newly discovered unspecific peroxygenase TanUPO from the ascomycetous fungus Truncatella angustata and several known UPOs (AaeUPO, CraUPO, MroUPO, MweUPO and CglUPO)to convert arachidonic acid (AA) to synthesize epoxyeicosatrienoic acid (EET) and hydroxyeicosatetraenoic acid (HETEs). It was found that TanUPO can convert more than 90% of AA to 14,15-EET, while the remaining UPO converted AA to another EET isomer or 18-/19-HETEs. Finally, the authors used AlphaFold2 to perform protein structure simulation and ligand docking of TanUPO to explain and illustrate the synthesis of single EET by TanUPO. This research has important implications for the biosynthesis of EET. Several key issues need to be discussed carefully for accept for publication in this journal.
Specific comments:
1. Please supplement HPLC-ELSD and HPLC-MS2 results for EET and HETE standards.
2. The size of the protein cannot be seen from the SDS-PAGE, and the protein bands were diffuse. The positions of the LMW Standards on the left and right are different. For example, the 35kDa on the left is significantly lower than the 35kDa on the right. Please redo and supplement the Western Blot data.
3. Why is the calculation of output in Table 1 the last percentage? How is this calculated? Please add it in the materials.
4. There are many errors in the manuscript that the chart numbers do not correspond, please check and revise the full text.
5. Please add an explanation why the production of EETs decreased at 120-180 minutes.
6. Why choose the 5th ranked model for follow-up research, lack of specific analysis, please supplement the data and conduct a comparative analysis.
7. “In all protein models, no solutions were found for C1 to C4, C7, C10, C13 and C16 to C18. However, fits were found for all carbons involved in double bonds as well as in C19 and C20 oxyfunctionalization (results not shown)”, Please supplement the fitted data in the manuscript.
8. Please discuss in depth the differences between the manuscript study and the B.megaterium study.
9. Lines 438-441 of the discussion cannot be directly observed in the picture, please provide the specific docking parameter comparison between TanUPO, MroUPO and AaeUPO and the substrate AA.
Author Response

(The authors gave the same response as above.)

Round 2
Reviewer 1 Report
The authors corrected and completed the manuscript in a satifying way.
Only minor points need to be corrected:
The use of Figure or Fig. is not homogeneous in the text.
Line 292, use upper case for table (Table).
Line 316, replace “Fig. S 8” by “Fig. S 18”.
Line 330, replace “SAC (substrate access channel)” by “substrate access channel (SAC)”.